# AdaptVis: Spatial Reasoning in Vision-Language Models Requires Adaptive Attention

## Abstract

Large Vision Language Models (VLMs) have long struggled with visual-centric understanding, with spatial reasoning emerging as a notable bottleneck due to its reliance on effective image processing. Surprisingly, even simple spatial reasoning tasks, such as recognizing "under" or "behind" relationships between only two objects, pose significant challenges for current VLMs. We believe it is crucial to use the lens of *mechanism interpretability*, diving into the model's internal states to examine the interactions between image and text tokens during spatial reasoning. Our analysis of attention behaviors reveals significant differences in how VLMs allocate attention to image tokens versus text tokens. By visualizing the areas of images that receive the highest attention scores throughout intermediate layers, we observe a notable pattern: errors often coincide with attention being misdirected towards irrelevant objects within the image. Moreover, such attention patterns exhibit substantial differences between familiar (e.g., "*on the left side of*") and unfamiliar (e.g., "*in front of*") spatial relationships. Motivated by these findings, we propose AdaptVis based on inference-time confidence scores to sharpen the attention on highly relevant regions when the model exhibits high confidence, while smoothing and broadening the attention window to consider a wider context when confidence is lower. This novel decoding method shows significant improvement (e.g., up to a 50 absolute point improvement) on spatial reasoning benchmarks such as WhatsUp and VSR with negligible additional cost.

## 1 Introduction

Despite rapid advancements in Large Vision-Language Models (VLMs), a significant deficiency persists, i.e., their struggle with vision-centric abilities (Gao et al., 2023; Kamath et al., 2023; Tong et al., 2024a; Chen et al., 2024a). This limitation is particularly notable in spatial reasoning given the simplicity of the task. Spatial reasoning involves inferring basic relationships between just two objects, such as "*left*", "*right*", "*above*", "*below*", "*behind*", or "*front*", as shown in Figure 1. For example, given the image with a book "*behind*" the candle in Figure 1, VLMs describe the book as being "*left*" of the candle. This error is not an isolated incident but a frequent recurring pattern that highlights a critical bottleneck in how VLMs process visual information.

Recent studies have started to probe potential issues in the vision-centric processing of VLMs, questioning whether vision encoders like CLIP (Radford et al., 2021) adequately capture visual information (Tong et al., 2024b;a). However, a crucial aspect remains underexplored: the intricate interplay between vision and text tokens within models' internal states. This gap underscores the need for diving deeper into mechanism in the geometric understanding of visual scenes. While recent work has made progress in the failure analysis of VLMs, investigating why VLMs make errors (Kamath et al., 2023) such as object hallucination, these studies have primarily focused on object-centric tasks to address the alignment between objects and the semantics of concepts. However, spatial reasoning presents a fundamentally different and more challenging problem that has yet to be thoroughly examined through the lens of mechanism interpretability. Spatial reasoning requires not only object identification but also the ability to localize within visual scenes and understand their interaction with text tokens. Therefore, we dive into the internal mechanisms of VLMs, specifically examining how vision and text tokens are processed and interact in spatial reasoning.

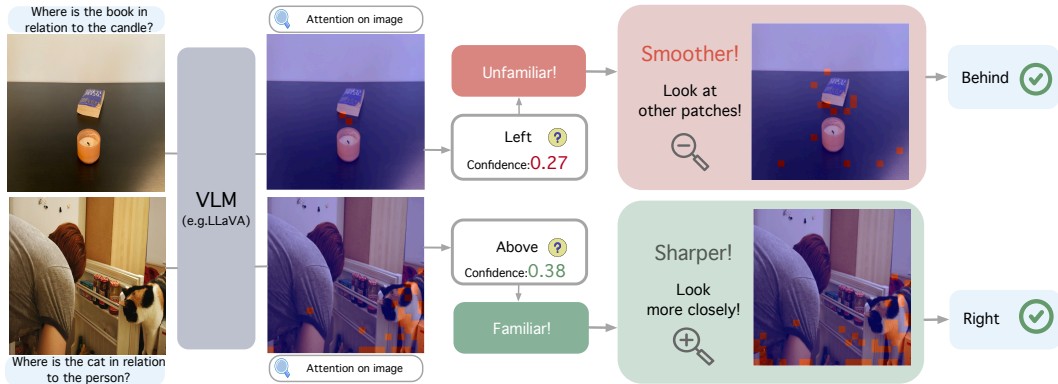

Figure 1: The framework of ADAPTVIS. We adaptively intervene in the temperature of the attention logits of the image tokens. 1). For generations with **high** confidence, we trust the attention pattern and **sharpen** the attention distribution. 2). For generations with **low** confidence, we **smoothen** the attention distribution to broaden the context window for better concentration on the correct objects.

We first hypothesize that if the VLM can adequately ground itself in visual scenes, it should be capable of localizing objects and understanding their geometric structures within visual scenes. Building on this, we further hypothesize that many failures of the VLM stem from how it distributes its attention over the image tokens. Our focus on attention behavior is motivated by the intuitive connection between attention mechanisms and spatial localization, particularly regarding how attention correlates with object localization and is influenced by textual tokens. By closely examining the model's attention scores(logits)[1] to images, we seek to answer the following questions: 1) **What is the potential cause for these failures?** 2) **Can we detect these failures within the model itself?** In other words, how do these failures manifest through internal patterns?

Our systematic analysis of attention scores throughout intermediate layers reveals a critical pattern: the model tends to "**focus**" on question-relevant entities in the middle to high intermediate layers when it could answer correctly, as evidenced by higher attention scores on these entities. Conversely, failures often correlate with the model attending to irrelevant objects, as shown in Figure 6. Moreover, we observe a pattern in the model's output behavior: it frequently predicts spatial relationships corresponding to the regions receiving the highest attention scores. For instance, if attention scores are higher towards the lower part of the image, the model is likely to output "under". Thus, incorrect outputs may result from model's failure to properly attend to the correct objects. Our analysis also reveals image tokens receive significantly less attention than text tokens. This supports our hypothesis that textual priors often outweigh visual information, hindering performance on vision-centric tasks. For example, when shown an image of a cup clearly "*under*" a table, VLMs often misinterpret it as being on the "*left*" side, likely due to over-reliance on common textual patterns over visual evidence.

As a result, our goal is to make VLMs look at images cleverly, i.e., sharpening its focus when confident, and broadening or shifting its focus window when the model doubts its predictions. While previous work has shown the effectiveness of guiding model attention using additional object labels (Chen et al., 2024a; Zhang et al., 2024) or external object detectors (Li et al., 2024; You et al., 2024), we leverage the signal within the model itself to regulate its attention behavior. We further examine several signals from the model's internal states to design an inner belief-based metric for guiding attention, and finally choose the model's generation probability as a reliable indicator of its confidence. When the model is confident with its own generation (Figure 1) (the probability of generated tokens exceeds a threshold), we trust the attention pattern on the image tokens by sharpening the attention distribution. On the other hand, when the model is not confident, we smooth the image attention distribution to expand the focused context window, thereby increasing the likelihood of focusing on relevant objects. We include the exploration of other potential inner belief-based metrics at Appendix.

Experiments are done on typical spatial reasoning benchmarks WhatsUp (Kamath et al., 2023) and VSR (Liu et al., 2023). To clearly observe and interpret model's inner working mechanism during spatial relationship generation, we adopt Question-Answering (QA) settings by reformatting these benchmarks. These benchmarks cover a wide range of image distributions including both

---

[1]We use "attention scores" to indicate the attention logits in our whole paper.

synthetic data (with clean backgrounds containing only two objects) to real scenario data (with noisy backgrounds and multiple objects). Our results show that ADAPTVIS achieves up to 50 absolute point gains across all benchmarks with minimal computational overhead.

## 2    BACKGROUND

We center our analysis on the attention behavior across layers to investigate how VLMs distribute their attention over image tokens, aiming to gain deeper insights into spatial reasoning errors.

**Notation**    Large VLMs like LLaVA (Liu et al., 2024a) consist of three components: a visual encoder like CLIP (Radford et al., 2021), a pretrained language model and a projector to connect these two parts. The visual encoder functions as a perception tool to "see" the image, while the image information is processed through a projector to be mapped into the token space. The LLM part is often based on the transformer architecture, consisting of $L$ layers stacked together. Each layer consists of two major components: a Multi-Head Attention (MHA) module, followed by a feed-forward network. For each layer $l$, given the input $\boldsymbol{X} \in \mathbb{R}^{n \times d}$ (where $n$ is the number of tokens and $d$ is the embedding dimension), the MHA module performs the self-attention function in each head $N_h$, and the output is a concatenation of all heads' outputs: $\text{MHA}^{(l)}(\boldsymbol{X}) = \text{Concat}\left(\boldsymbol{N}_h^{(l,1)}, \ldots, \boldsymbol{N}_h^{(l,H)}\right) \boldsymbol{W}_o$, where $H$ is the number of heads, $\boldsymbol{N}_h$ is the output of head $N_h$, computed as

$$\boldsymbol{N}_h^{(l,h)} = \text{Softmax}(\boldsymbol{A}^{(l,h)})\boldsymbol{V} = \text{Softmax}\left(\frac{\boldsymbol{Q}\boldsymbol{K}^\top}{\sqrt{d_h}} + \boldsymbol{M}\right)\boldsymbol{V}, \tag{1}$$

where attention logits $\boldsymbol{A}^{(l,h)}$ is computed via $\boldsymbol{Q} = \boldsymbol{X}\boldsymbol{W}_{q_h}, \boldsymbol{K} = \boldsymbol{X}\boldsymbol{W}_{k_h}, \boldsymbol{V} = \boldsymbol{X}\boldsymbol{W}_{v_h}$, and $\boldsymbol{W}_{q_h}, \boldsymbol{W}_{k_h}, \boldsymbol{W}_{v_h} \in \mathbb{R}^{d \times d_h}$ are learnable projection matrix of the head $N_h$. A causal mask $\boldsymbol{M}_{ij} = 0$ if $i \geq j$, and $-\infty$ otherwise, prevents tokens from attending to future tokens.

## 3    DIVING INTO ATTENTION DISTRIBUTIONS

In this section, we systematically examine the influence of the absolute values of attention logits over image tokens during spatial reasoning.

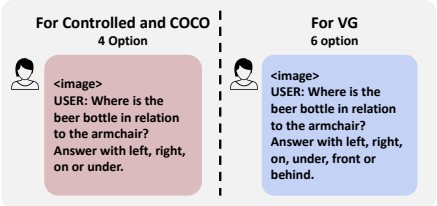

Figure 2: Prompts used in our evaluation.

**Experiment settings**    We select a widely-used spatial reasoning benchmark WhatsUp (Kamath et al., 2023) since it contains both synthetic data and realistic data. The synthetic data (**Controlled_Image**)[2] features clean backgrounds with two objects, as shown in the upper example of Figure 1. It comprises two subsets: **Controlled_A**, with one large object (e.g., table) and one small object (e.g., cup), and **Controlled_B** with two small objects (e.g., book and plate). The realistic data, as shown in the lower image of Figure 1, contains complex backgrounds with multiple objects, sourced from MS COCO (Lin et al., 2014) and Visual Genome (Krishna et al., 2017) (referred to as **COCO** and **VG** later). While realistic images present more challenges in localization, the synthetic data enables clearer observation of VLMs's inner workings with just two objects. Each image is paired with a ground truth caption describing the spatial relationship of two objects. We reformat the original ⟨*image, caption*⟩ setting into a generative question-answering setting ⟨*image, question, spatial_label*⟩, enabling evaluation of generative models like VLMs and tracing of internal states. Questions are generated using GPT-4 (OpenAI, 2024), with prompts shown in Figure 2.

For evaluation, we apply **accuracy** of exact match as the primary metric. To maintain consistency in the label space across datasets, we use a four-option setting ⟨*left, right, on, under*⟩ for the Controlled_Image and COCO subsets, and a six-option setting ⟨*left, right, on, under, behind, front*⟩ for the VG as it contains additional spatial annotations.

Another aspect we analyze is the label distribution. In **Controlled_Image**, labels are uniformly distributed across categories (e.g., equal number of samples for "*left*", "*right*", "*on*", "*under*"). Another interesting feature of this dataset, which contributes to our choice to use it, is its contrastive

---

[2]We use Cont_A and Cont_B as abbreviations in the paper.

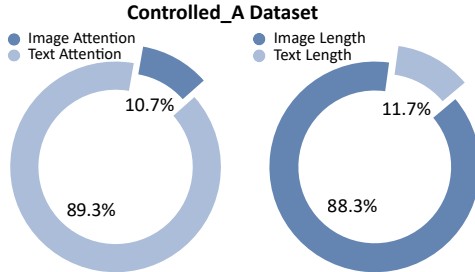 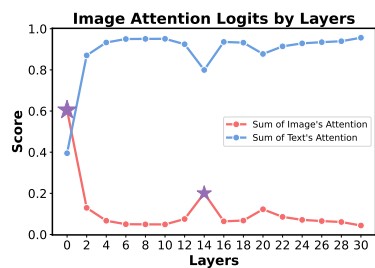

Figure 3: The comparison of the average attention scores received and the length between the image tokens and text tokens in Cont_A.

Figure 4: The variance of image token's attention scores through the layers in Cont_A benchmark.

setting. Controlled_Image includes **pairs** of images with same objects in both "*left*" and "*right*" positions, and **sets** of same objects exhibiting "*left*", "*right*", "*on*", and "*under*" relationships. It enables us further assess model performance using **pair accuracy** and **set accuracy**, requiring correct identification of all relationships within a pair or set, defined by Kamath et al. (2023). It provides a comprehensive evaluation of spatial relationships, especially when the objects involved are identical.

## 3.1 VLMs ALLOCATE SPARSE ATTENTION TO THE IMAGE

We analyze how output tokens attend to image tokens by extracting the attention logits across layers, and present the following key findings: 1) The sum of attention scores to **the image tokens is significantly lower** than that to all the input text tokens, despite the considerably higher number of image tokens. In Figure 3, we focus on the attention scores from the first generated token and sum the attention allocated to the image tokens. The results reveal that image tokens receive substantially less attention, with text tokens receiving approximately nine times more. Although the image sequence has a length of 576, compared to the text sequence, which typically ranges from 30 to 40 tokens in our short question-answering setting, the model predominantly focuses on text when generating outputs. In other words, image information is sparsely processed by the language model.

To further investigate, we extract attention scores across the middle layers to observe the flow of information through the model. As shown in Figure 4, attention to the input image is highest in the initial layers but decreases sharply within the first two layers to a very low value. In the middle layers, we observe a slight increase, with attention reaching a modest peak. This leads to our second finding: 2) **the model processes the image information primarily through the intermediate layers**. Similar observations have been made by Halawi et al. (2023) and Geva et al. (2023), where intermediate layers tend to encode more factual knowledge.

## 3.2 IS THE ANSWER MORE ACCURATE IF THE MODEL SEES THE IMAGE MORE?

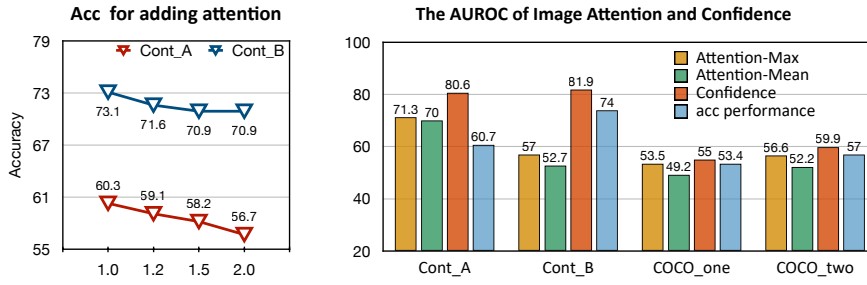

Figure 5: Left: Accuracy for adding image attention for all image tokens. Right: AUROC of attention scores relative to the model's confidence.

Building on earlier observations, a natural question arises: since the attention scores assigned to the image are relatively low compared to those for text, could increasing attention to the image improve the factual accuracy of the model? To investigate this, we conduct a calibration experiment using two statistical approaches. The first approach sums the attention scores assigned to the image as a metric, while the second extracts the highest attention score among the image tokens. These metrics are then evaluated to assess their effectiveness in distinguishing between correct and incorrect generations.

However, as shown in Figure 5, the AUROC score using attention (the yellow and green bar) is consistently lower than the AUROC score of the model's self-confidence (the yellow bar), which we measure by the probability of the output tokens. Additionally, we observe that the maximum attention score provides better calibration than the average attention score, suggesting that key information aligns more closely with maximal attention values. This suggests that the assumption "*the more attention the model pays to the image, the more accurate the results*" holds only partially true. This observation motivated us to explore more intelligent ways to focus on key visual features. In response, we conducted an additional experiment by incrementally increasing the attention weights across the entire image, where we intervened by augmenting the image attention logits with a constant coefficient, as described by Zhang et al. (2023). In Figure 5, we observe that adding a constant weight uniformly across all image tokens does not improve performance on spatial reasoning tasks.

## 4 DIVE INTO THE VISUAL PATTERNS

Building on our previous findings, it is clear that the overall attention allocated to the image is notably low, and the attention weights on the image tokens are insufficient for distinguishing the factuality of the model's outputs. These findings necessitate a more fine-grained analysis of this issue. To address this, we conduct a detailed investigation by mapping the 576 image tokens in LLaVA 1.5 to their corresponding image patches ($24 \times 24$). This visualization enables us to examine the attention patterns with greater granularity and clarity.

### 4.1 THE MODEL AUTOMATICALLY FOCUSES ON THE RELEVANT ENTITY WHEN CORRECTLY ANSWERING QUESTIONS.

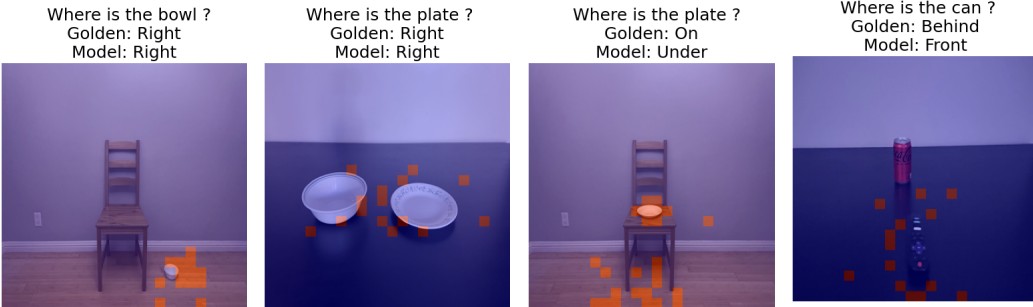

Figure 6: Attention visualization examples from the WhatsUp Dataset are shown. The left two examples are answered correctly, while the right two are incorrect. For correctly answered questions, the attention scores are precisely focused on the core entities mentioned. In contrast, incorrect answers show attention scores distributed to irrelevant image regions. The visualizations use attention from the 17th layer, and the title in each image is an abbreviation of "Where is A in relation to B".

We separately examine the attention patterns for correctly and incorrectly answered questions. Our observations reveal that hallucinations frequently occur due to two types of attention failures: (1) insufficient attention to the correct object, and (2) misplaced attention on irrelevant objects in the image. Figure 6 illustrates these findings. In the two correct examples on the left, the attention scores are well-aligned with the referenced entities, with sufficient focus. On the other hand, the two incorrect examples on the right demonstrate how the model incorrectly assigns attention, effectively "seeing" the wrong parts of the image. While these examples highlight the qualitative differences in attention patterns, they do not provide a quantitative metric that aligns with our goal–developing a method to detect the reliability of internal states and enable intervention. Therefore, the primary challenge lies in devising effective strategies to adjust the attention scores intelligently, given that we have no prior information about the scores until a single inference run generates the attention map.

### 4.2 WHEN CAN WE TRUST A MODEL'S ATTENTION PATTERN, AND WHEN CAN WE NOT?

To determine when to focus more on the image and when to focus more on the text, we begin by analyzing existing datasets to identify potential patterns. Our investigation starts with an examination of label distributions across different subsets of the WhatsUp dataset. As shown in Figure 7 (left), there is a clear label imbalance in the real-image datasets including COCO_two and VG_two, i.e., only

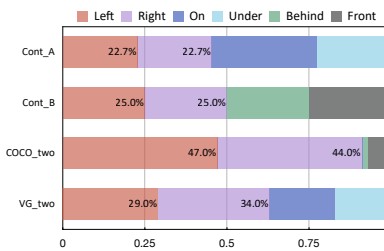 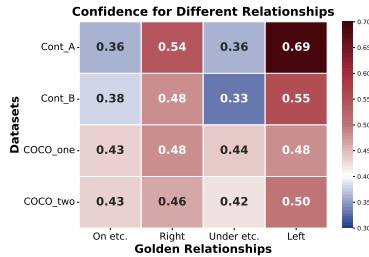

Figure 7: Left: Comparison of label distributions across different subsets. Right: The model's average confidence in different golden spatial relationships within WhatsUp. "On etc" includes "*on, above, top*", while "Under etc" includes "*under, below, bottom*" The red box highlights instances where the model is confident in its generation, while the blue box indicates the opposite.

a small portion of the samples have the relation of "*behind*" and "*front*". In contrast, the synthetic datasets, Cont_A and Cont_B, which are carefully curated, display more balanced label distributions. Additionally, we evaluate the model's confidence scores across these spatial relations by following the approach of Kadavath et al. (2022), where the probability of the generated output is used to compute confidence. Figure 7 (right) reveals that the model struggles with specific spatial relationships, such as "*on*" and "*under*", while demonstrating higher confidence in recognizing more common relationships, like "*left*" and "*right*". This observation aligns with our intuition that the model tends to show greater confidence in cases where it performs well and lower confidence where it struggles. This finding is consistent with previous work showing that models can convey their uncertainty through confidence scores (Kadavath et al., 2022; Xiong et al., 2024). Motivated by this, we propose using confidence as a metric to gauge the model's familiarity with spatial relationships in images.

## 5 ADAPTVIS

Motivated by our observation of attention misallocation causing spatial reasoning errors (Section 5.1) and the model's familiarity with spatial relationships (Section 5.2), we propose a novel decoding method, ADAPTVIS, and its basic version SCALINGVIS.

### 5.1 SCALINGVIS: TEMPERATURE SCALING TO IMAGE ATTENTION DISTRIBUTION

Our observations from Section 4.1 reveal that the model often misallocates attention logits within images, leading to errors in spatial reasoning. To mitigate this, we aim to improve the model's focus on key visual features, enhancing its ability to correctly ground spatial relationships, particularly in complex or ambiguous scenarios. To this end, we propose a simple yet effective approach that dynamically adjusts image attention by modifying the temperature of the attention distributions. Specifically, we intervene in the attention of the final input token ($n$-th position) to the image tokens.

$$\boldsymbol{A}_{n,j}^{(l,h)} = \begin{cases} \alpha \boldsymbol{A}_{n,j}^{(l,h)} & \text{if } j \in \mathcal{I} \\ \boldsymbol{A}_{n,j}^{(l,h)} & \text{otherwise} \end{cases} \quad (2)$$

where $\mathcal{I}$ represents the indices of all image tokens. In essence, we intervene in the attention score to the image tokens by multiplying a coefficient $\alpha$. We uniformly apply this coefficient to all $H$ heads across all $L$ layers to avoid the need for extensive hyperparameter search. Multiplying the attention scores in logit space by a coefficient $\alpha$ is equivalent to modifying the temperature $T$ for Softmax, where increasing $\alpha$ effectively decreases the temperature, leading to a sharper probability distribution among the image tokens.

**Experiment Setting** We select two widely-used benchmarks on evaluating the model's ability on spatial reasoning WhatsUp (Kamath et al., 2023) (introduced in Section 2), and VSR (Liu et al., 2023), which contains contains 1223 image-caption pairs with boolean labels. The original VSR is designed in ⟨*image*, *caption*⟩ format to evaluate encoder models without generation capabilities. To adapt it for our purposes, we utilize GPT-4o to generate questions for the VSR dataset. For evaluation, we report both accuracy and $F_1$ scores. A small validation set is allocated for each subset to optimize the temperature based on validation performance, and the final test is conducted on the test set. For both methods, the hyperparameter $\alpha$ is selected from $[0.5, 0.8, 1.2, 1.5, 2.0]$.

| Model | Cont_A | | | Cont_B | | | COCO_one | COCO_two | VQ_one | VQ_two |
|---|---|---|---|---|---|---|---|---|---|---|
| LLaVA-1.5 | 60.3 | 40.6 | 0.0 | 73.1 | 41.6 | 3.7 | 53 | 58.2 | 35.9 | 40.8 |
| +ScalingVis | 64.5 ↑4.2 | 40.6 ↑0.0 | 0.0 ↑0.6 | 75.2 ↑2.1 | 44.6 ↑3.0 | 9.8 ↑6.1 | 53.6 ↑0.6 | 59.4 ↑1.2 | 42.7 ↑6.8 | 48.1 ↑7.3 |
| Best $\alpha$ | 0.8 | | | 0.8 | | | 1.2 | 1.2 | 2.0 | 2.0 |
| LLaVA-1.6 | 59.7 | 41.8 | 31.6 | 63.0 | 39.1 | 3.7 | 59.7 | 41.8 | 31.6 | 7.3 |
| +ScalingVis | 97.0 ↑37.3 | 76.4 ↑34.6 | 54.5 ↑22.9 | 73.4 ↑10.4 | 48.9 ↑9.8 | 15.9 ↑12.2 | 63.1 ↑3.4 | 47.7 ↑5.9 | 38.2 ↑6.6 | 14.6 ↑7.3 |
| Best $\alpha$ | 0.8 | | | 2.0 | | | 1.2 | 1.5 | 2.0 | 2.0 |

Table 1: Accuracy on WhatsUp (Metrics are in $\times 10^{-2}$). Best-performing method per model and dataset are highlighted in bold; arrows indicate improvement over greedy decoding.

**Results** Our results for ScalingVis are presented in Table 1. By controlling the distribution of attention weights, spatial reasoning performance improves significantly, with gains of up to 37.2 absolute points. An interesting pattern emerges: a temperature below one tends to enhance performance on synthetic data in most cases (3 out of 4), while a temperature above one benefits real image datasets across all cases. Table 1 indicates that for **synthetic data**, smoothing the image attention logits improves performance. Conversely, for **real image datasets** (COCO and VG), the optimal temperature is consistently above one, demonstrating that a sharper attention distribution helps the language model discern relationships more effectively.

## 5.2 ADAPTVIS: CONFIDENCE-AWARE TEMPERATURE SCALING

One of the main limitations of SCALINGVIS is the unclear underlying mechanism behind how different values of $\alpha$ affect various distributions. Specifically, it remains unclear why synthetic data requires a lower temperature while real data benefits from a higher temperature. To address this, we aim to adaptively select the temperature on a per-sample basis. In this section, we introduce an extension of SCALINGVIS: ADAPTVIS Decoding.

$$\mathbf{A}_{n,j}^{(l,h)} = \begin{cases} \alpha_1 \mathbf{A}_{n,j}^{(l,h)} & \text{if } j \in \mathcal{I} \\ \mathbf{A}_{n,j}^{(l,h)} & \text{otherwise} \end{cases}, \text{if } \mathcal{C} < \beta \qquad \mathbf{A}_{n,j}^{(l,h)} = \begin{cases} \alpha_2 \mathbf{A}_{n,j}^{(l,h)} & \text{if } j \in \mathcal{I} \\ \mathbf{A}_{n,j}^{(l,h)} & \text{otherwise} \end{cases}, \text{if } \mathcal{C} > \beta$$

$$(3a) \qquad\qquad\qquad\qquad (3b)$$

**Confidence-Based Attention Intervention** Recall from Section 4.2 that we observe two distinct patterns in the model's factuality behavior: (1) synthetic data presents more unfamiliar cases than the real data, and (2) VLMs could express uncertainty through confidence scores. These insights motivate using confidence scores as a metric for adaptive intervention in the model's internal states. Our intuition is straightforward: when confidence is low, suggesting that the attention pattern may be unreliable, we smooth the attention distribution. This encourages the model to explore a broader range of image regions, increasing the likelihood of focusing on the correct patches. Conversely, when confidence is high and attention is dispersed across the image, we sharpen the distribution to concentrate on key objects more effectively. Specifically, we apply the targeted intervention to the

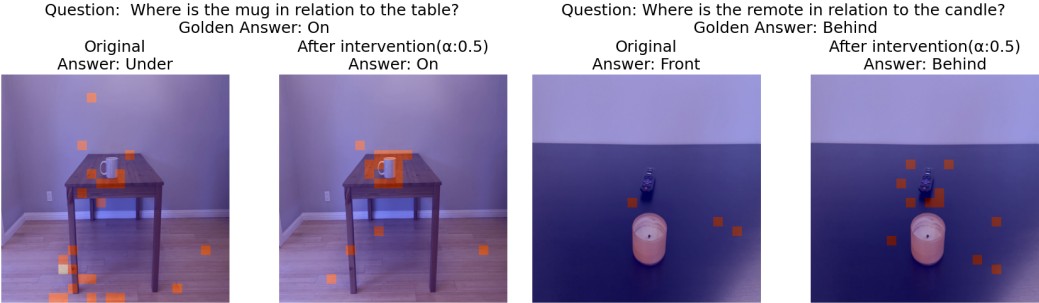

Figure 8: Attention scores on the patches before and after our intervention in the 17th layer for the images in **synthetic** datasets Cont_A and Cont_B. We employ a "**smoothing**" intervention method to **expand the context length** of the model's focused area. From the figure, it is evident that the model's focused position undergoes significant changes after our intervention.

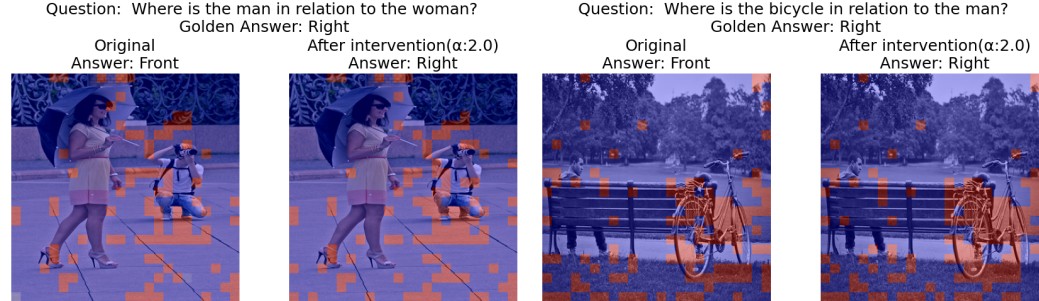

Figure 9: Attention scores on the patches before and after our intervention in the 17th layer for the images in **real** datasets COCO and VG. We utilize a "**sharpening**" intervention method to enhance the original attention pattern. The highlighted areas remain largely consistent, with our method serving to reinforce the focus rather than significantly altering it.

attention of the last input token (at the $n$-th position) directed toward the image tokens as shown in Equation 5.2. Overall, we use (1) a large $\alpha > 1$ when the Confidence $\mathcal{C}$ is large. This sharpens the attention distribution and the relevant objects are paid more attention to. (2) a small $\alpha < 1$ when the confidence $\mathcal{C}$ is small. This mitigates the model's excessive concentration on certain image tokens and makes the overall attention distribution smoother across the image.

To illustrate the impact of coefficients greater or less than 1 on different relationships, Figure 10 shows accuracy and confidence variance for various ground-truth relationships. Results indicate that for familiar relationships like "left" (red) and "right" (blue), coefficients greater than 1 boost accuracy and confidence. Conversely, for less familiar relationships like "under" (green) and "on" (yellow), coefficients less than 1 improve accuracy and confidence.

# 6 EXPERIMENTS

## 6.1 EVALUATION SETTINGS

We use the same hyperparameters for SCALINGVIS as in Section 5.1. For ADAPTVIS, we optimize $\alpha_1$, $\alpha_2$, and $\beta$ using the validation set from each distribution. Notably, we find that model performance is robust across a range of these hyperparameters, generalizing effectively to other subsets within the same distribution (as demonstrated in Table 4). We maintain consistency by using the same range of $\alpha$ values for both methods. For $\beta$, we adjust per dataset: for WhatsUP, we select values from $[0.3, 0.65]$ for LLaVA-1.6 and $[0.2, 0.55]$ for LLaVA-1.5 with a grid size of $0.05$ (this higher range is due to LLaVA-1.6 generally exhibiting higher confidence than LLaVA-1.5); for VSR, we take the mean value of the average confidence scores corresponding to the two labels.

## 6.2 RESULTS

Our main results are presented in Table 2. By controlling the distribution of attention weights, we observe a significant improvement in spatial reasoning ability, with gains of up to 50 absolute points. In most cases, ADAPTVISachieves the best performance, particularly for synthetic datasets, as shown in Table 3, where it significantly outperforms the generalized method. These findings suggest that model performance varies considerably with the label distribution of the dataset, and smoothing the distribution (by applying a coefficient smaller than 1) enhances performance. For the real-image

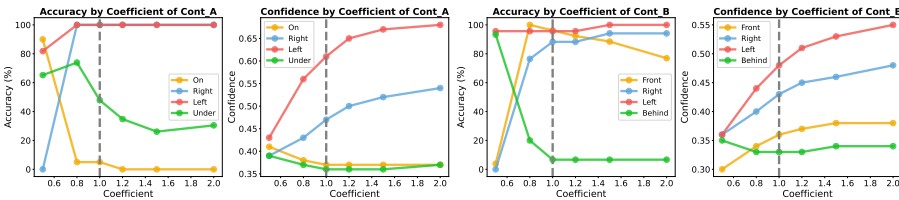

Figure 10: Accuracy and Confidence comparison for different coefficients $\alpha$.

| Model | What's Up | | | | VSR | |
|---|---|---|---|---|---|---|
| | COCO_one | COCO_two | VQ_one | VQ_two | Exact Match | F1 score |
| LLaVA-1.5 | 53.0 | 58.2 | 35.9 | 40.8 | 62.4 | 51.3 |
| +VCD | 53.3 ↑0.3 | 58.2 | 35.8 ↓0.1 | 42.5 ↑1.7 | 62.4 | 50.6 ↓0.7 |
| +Dola | **53.7 ↑0.7** | 57.5 ↓0.7 | 36.2 ↑0.3 | 42.1 ↑1.3 | 62.8 ↑0.4 | 53.2 ↑1.9 |
| +SCALINGVIS | 53.6 ↑0.6 | 59.4 ↑1.2 | 42.7 ↑6.8 | 48.1 ↑7.3 | 64.9 ↑2.5 | 62.5 ↑11.2 |
| +ADAPTVIS | 53.6 ↑0.6 | **59.9 ↑1.7** | **42.7 ↑6.8** | **48.1 ↑7.3** | **65.0 ↑2.6** | **62.5 ↑11.2** |
| LLaVA-1.6 | 59.7 | 41.8 | 31.6 | 7.3 | 58.8 | 29.4 |
| +VCD | 60.6 ↑0.9 | 44.9 ↑3.1 | 33.8 ↑2.2 | 11.6 ↑4.3 | 58.8 | 29.4 |
| +Dola | 59.7 | 41.5 ↓0.3 | 31.5 ↓0.1 | 7.3 | 59.3 ↑0.5 | 31.2 ↑1.8 |
| +SCALINGVIS | 63.1 ↑3.4 | **47.7 ↑5.9** | **38.2 ↑6.6** | 14.6 ↑7.3 | 59.1 ↑0.3 | 30.6 ↑1.2 |
| +ADAPTVIS | 63.1 ↑3.4 | 47.7 ↑5.9 | 35.2 ↑3.6 | **17.2 ↑9.9** | **62.7 ↑3.9** | **39.3 ↑9.9** |

Table 2: Results on WhatsUp and VSR (Metrics are in $\times 10^{-2}$). Best-performing method per model and dataset are highlighted in bold; arrows indicate improvement over greedy decoding.

| Model | Controlled_A | | | Controlled_B | | |
|---|---|---|---|---|---|---|
| | Acc | Pair Acc | Set Acc | Acc | Pair Acc | Set Acc |
| LLaVA-1.5 | 60.3 | 40.6 | 0.0 | 73.1 | 41.6 | 3.7 |
| +VCD | 61.5 ↑1.2 | 39.4 ↓1.2 | 0.0 | 73.4 ↑0.3 | 42.2 ↑0.6 | 3.7 |
| +Dola | 61.2 ↑0.9 | 41.6 ↑1.0 | 0.0 | 73.4 ↑0.3 | 42.2 ↑0.6 | 3.7 |
| +SCALINGVIS | 64.5 ↑4.2 | 40.6 | 0.0 | 75.2 ↑2.1 | 44.6 ↑3.0 | 9.8 ↑6.1 |
| +ADAPTVIS | **84.9 ↑24.6** | **61.2 ↑20.6** | **30.3 ↑30.3** | **83.8 ↑10.7** | **55.7 ↑14.1** | **18.3 ↑14.6** |
| LLaVA-1.6 | 48.2 | 37.6 | 0.0 | 63.0 | 39.1 | 3.7 |
| +VCD | 61.8 ↑13.6 | 41.8 ↑4.2 | 10.9 ↑10.9 | 65.4 ↑2.4 | 41.6 ↑2.5 | 7.3 ↑3.6 |
| +Dola | 48.2 | 37.6 | 0.0 | 62.7 ↓0.3 | 39.1 | 3.7 |
| +SCALINGVIS | 97.0 ↑48.8 | 76.4 ↑38.8 | 54.5 ↑54.5 | 73.4 ↑10.4 | 48.9 ↑9.8 | 15.9 ↑12.2 |
| +ADAPTVIS | **98.2 ↑50.0** | **78.8 ↑41.2** | **57.0 ↑57.0** | **73.4 ↑10.4** | **48.9 ↑9.8** | **15.9 ↑12.2** |

Table 3: Results on WhatsUp's Cont_A and Cont_B. (Metrics in $\times 10^{-2}$). Best-performing method per model and dataset are highlighted in bold; arrows indicate improvement over greedy decoding.

datasets, as shown in Table 2, the adaptive method performs slightly better than the generalized approach, indicating the model's robustness across different label distributions.

It is important to note that the LLaVA-series models are trained on the COCO dataset, which makes them highly confident and familiar with COCO and VG image types. Hence trusting the model's self-belief and sharpening image attention improves performance. Notably, for datasets containing a high proportion of unfamiliar images, the adaptive setting proves to be significantly more effective.

**Reverse curse?** Additionally, we observe that the model exhibits a "reverse curse" phenomenon similar to that has been seen in language models (Berglund et al., 2023). When we reverse the order of the entities in Cont_A (e.g., asking the model, "*Where is the armchair in relation to the beer bottle?*" instead of "*Where is the beer bottle in relation to the armchair?*"), there is a significant drop in performance. As shown in Figure 11, the model's performance declines dramatically from a high score to an exceptionally low one. This reveals that the existing VLMs's attention pattern and generation results could be significantly impacted by the prompt. Our methods, however, consistently improve the model's performance. It suggests that adaptively intervening the attention score is a generalizable method for different prompts.

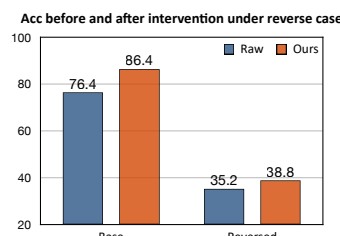

Figure 11: Performance comparison before and after SCALINGVIS intervention ($\alpha = 0.5$).

**Hyperparameter Out-of-Domain Test** We evaluated the generalizability of common hyperparameters across datasets. Specifically, we applied the same set of

four-option prompts to Controlled_Images and COCO subsets. The results in Table 4 indicate that ADAPTVIS consistently performs well across all subsets, confirming its generalizability.

| Model | Cont_A | | | | Cont_B | | | COCO_one | COCO_two |
|---|---|---|---|---|---|---|---|---|---|
| | Acc | Pair Acc | Set Acc | Acc | | Pair Acc | Set Acc | Acc | Acc |
| LLaVA-1.5 | 60.3 | 40.6 | 0.0 | 73.1 | | 41.6 | 3.7 | 53.0 | 58.2 |
| +Ours | 60.3 | 41.8 ↑1.2 | 2.4 ↑2.4 | 76.5 ↑3.4 | | 48.3 ↑6.7 | 13.5 ↑9.8 | 53.6 ↑0.6 | 59.4 ↑1.2 |
| Best $\alpha$ | | | | $\alpha_1 = 0.5$ | $\alpha_2 = 1.2$ | $\beta = 0.3$ | | | |

Table 4: OOD test results on WhatsUp (Metrics in $\times 10^{-2}$). Arrows show growth over baseline.

**How do the absolute values of attention scores vary before and after intervention?** Figure 12 visualizes the attention logits for two-option Cont_A ($\alpha$=0.5) and six-option VG ($\alpha$=2). An $\alpha$ of 0.5 decreases the absolute value of logits across layers, while an $\alpha$ of 2 increases them as layers progress. This indicates that an $\alpha$ larger than 1 could strengthen the model's orginal attention pattern.

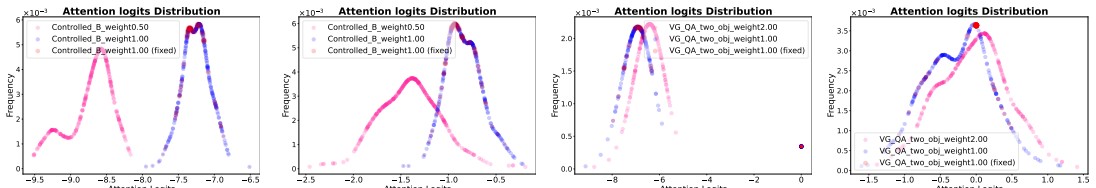

Figure 12: Attention logit distribution before (blue) and after (pink) intervention in the 14th layer (a randomly chosen middle layer). From left to right, the plots represent: mean and max attention values across heads for Cont_A, and mean and max attention for Cont_B. Red dots mark cases corrected by our intervention. We could see that $\alpha$ of 0.5 shifts the line left, while $\alpha$ of 2 shifts it right.

## 7 RELATED WORK

The first line of our related work focuses on research into attention patterns in language models. Some studies on attention patterns in LLMs reveal biased attention across context windows, such as ineffective use of the middle context (Liu et al., 2024b) and initial token attention sinks (Xiao et al., 2023). While some approaches use fine-tuning to overcome these biases (An et al., 2024), training-free methods like input-adaptive calibration (Yu et al., 2024b) and position-specific interventions (Yu et al., 2024a) offer efficient alternatives. PASTA (Zhang et al., 2023), a closely related method, emphasizes attention on selected segments for specific heads; we extend this to VLMs without manual segment specification or multiple validation runs. Our work is also related with failure analysis in VLMs, VLMs have been shown to hallucinate more in multi-object recognition tasks and rely on spurious correlations (Chen et al., 2024c), with systematic visual limitations highlighted from a CLIP perspective (Tong et al., 2024b). Our work also connects to the decoding strategies for reducing hallucinations decoding strategies to mitigate hallucinations include contrastive decoding focusing on image regions (Leng et al., 2024), preference tuning through data augmentation (Wang et al., 2024), and methods leveraging contrastive layers for enhanced knowledge extraction (Chuang et al., 2023), as well as activation-based optimal answer identification (Chen et al., 2024b).

## 8 CONCLUSION AND FUTURE WORK

Our research uncovers the inner working mechanism of VLMs during spatial reasoning, which is a critical limitation in VLMs and constrains their practical utility when requiring geometric understanding of visual scenes. We identify key insights through an in-depth study of attention behaviors across layers: 1) VLMs allocate surprisingly insufficient attention to image tokens; 2) the location of attention on image tokens is more crucial than quantity; and 3) generation confidence serves as a reliable indicator of its familiarity with the image and the correctness of its attention pattern. Based on these findings, we propose ADAPTVIS, a novel decoding method that dynamically adjusts attention distribution, significantly improving spatial reasoning performance. Future research could focus on further exploring mechanism interpretability of VLMs on complicated geometric structure understanding, such as long-horizon spatial reasoning, and investigate other reasons for spatial reasoning bottleneck, such as the memorization of training data.

## REPRODUCIBILITY STATEMENT

To ensure the reproducibility of our ADAPTVIS approach, we have included all necessary hyperparameters and training configurations in Section 6. Our source code, including model architecture details and pre-processing scripts, will be made available as an anonymous downloadable link in the supplementary materials. Furthermore, for datasets used in our experiments, a detailed description of the data preparation, along with any custom transformations, is described in Section 3. This structured documentation aims to allow for seamless replication of our results across different environments.

## ETHICS STATEMENT

This paper presents a method to enhance spatial understanding of visual-language models. We use publicly available datasets and models that may contain biases. However, we think that there are no ethical concerns that need to be highlighted at the current moment. As we enhance the spatial reasoning capabilities of VLMs, we must also consider the ethical implications. Improved spatial understanding in AI systems could raise privacy concerns, particularly in surveillance applications. It is crucial that the development of these technologies is accompanied by robust ethical guidelines and privacy safeguards. Furthermore, as these models become more capable, it is important to ensure equitable access to the benefits they provide, avoiding scenarios where advanced AI widens existing societal disparities.

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

# A APPENDIX

## A.1 LIMITATIONS

Firstly, our methods, SCALINGVIS and ADAPTVIS, specifically address model-related spatial hallucinations and self-alignment issues but are not designed to handle errors outside the language model's capabilities, such as the CLIP failures discussed by Tong et al. (2024b). Secondly, ADAPTVIS relies on distribution-based confidence to adaptively set the confidence threshold $\beta$, we also observe that the optimal $\alpha$ and $\beta$ is different across different distributions and prompts. This dependence on a validation set for tuning poses a limitation on its applicability.

## A.2 BROAD IMPACT

Our research into the spatial reasoning capabilities of Large Vision Language Models (VLMs) has significant implications across various domains of artificial intelligence and its real-world applications. First and foremost, our findings highlight a critical limitation in current VLMs: while they excel at object recognition, they struggle with basic spatial relationships. This gap between recognition and spatial understanding has far-reaching consequences for the practical deployment of VLMs in scenarios requiring geometric comprehension of visual scenes. Industries such as robotics, autonomous navigation, and assistive technologies for the visually impaired are particularly affected. For instance, a robot that can identify objects but cannot understand their spatial relationships may struggle with tasks like picking and placing items or navigating complex environments. Similarly, autonomous vehicles might face challenges in interpreting traffic scenarios accurately, potentially compromising safety.

Our development of ADAPTVIS, a novel decoding method that dynamically adjusts attention distribution based on the model's confidence, represents a significant step forward. By enhancing VLMs' performance on spatial reasoning tasks, ADAPTVIS could unlock new possibilities in various fields. In healthcare, improved spatial reasoning could lead to more accurate interpretation of medical imaging, potentially improving diagnostic accuracy. In augmented reality applications, better spatial understanding could enable more immersive and interactive experiences. For assistive technologies, enhanced spatial reasoning could provide more accurate and useful descriptions of environments to visually impaired individuals, significantly improving their independence and quality of life.

Looking ahead, our work opens up new avenues for research in AI and cognitive science. The exploration of mechanism interpretability in VLMs, particularly for complex geometric structures and long-horizon spatial reasoning, could provide insights into how artificial systems process and understand spatial information. This could not only advance AI capabilities but also contribute to our understanding of human spatial cognition. Additionally, investigating the role of training data memorization in spatial reasoning bottlenecks could lead to more efficient and effective training methods for future AI models.

In conclusion, our research not only addresses a fundamental limitation in current VLMs but also paves the way for more versatile and capable AI systems. As we continue to advance VLMs' capabilities in visually-driven tasks requiring nuanced spatial understanding, we have the potential to significantly impact various sectors of society, from healthcare and assistive technologies to urban planning and environmental monitoring. The future research ahead in this field is both exciting and challenging, requiring ongoing collaboration between researchers, ethicists, and policymakers to ensure that these advancements benefit society as a whole.

## A.3 PROMPT SENSITIVITY ANALYSIS

To assess the robustness of our method, we varied the number of options in prompts for specific WhatsUp subsets. For Cont_A and Cont_B, we reduce the number of options to two, simplifying the task. Conversely, for COCO subsets, we increase the options to six, making it more challenging. Table 5 shows that our SCALINGVIS method maintains consistent performance across all cases, demonstrating robustness.

| Model | Cont_A | | | Cont_B | | | COCO_one | COCO_two |
|---|---|---|---|---|---|---|---|---|
| | Acc | Pair Acc | Set Acc | Acc | Pair Acc | Set Acc | Acc | Acc |
| LLaVA-1.5 | 76.4 | 43.0 | 4.8 | 74.6 | 41.0 | 1.2 | 30.8 | 42.6 |
| +Ours | 86.4 ↑10.0 | 61.2 ↑18.2 | 27.9 ↑23.1 | 87.8 ↑13.2 | 59.3 ↑18.3 | 22.0 ↑20.8 | 36.0 ↑5.2 | 48.6 ↑7.3 |
| Best $\alpha$ | | 0.5 | | | 0.5 | | 2 | 2 |

Table 5: Results on WhatsUp (Metrics in $\times 10^{-2}$). Arrows show improvement over greedy decoding.

### A.4 RELATED WORK

**Attention Patterns in Language Models**   Ongoing research has shown how large language models (LLMs) exhibit biased attention across different parts of the context window. Liu et al. (2024b) find that LLMs fail to effectively utilize the information in the middle of a long context window. Meanwhile, Xiao et al. (2023) reveals an attention sink at the initial tokens of the input. Besides finetuning methods to overcome such biases (An et al., 2024), some training-free methods have been proposed with the benefit of their efficiency. Yu et al. (2024b) proposes to use input-adaptive calibration to adjust the attention scores, while Yu et al. (2024a) intervenes in position-specific hidden dimensions to alleviate the lost-in-the-middle phenomenon. A closely related work to ours is PASTA (Zhang et al., 2023), which emphasizes the attention scores of specific text segments for selected attention heads. We further develop this motivation on vision language models. Moreover, our method does not require a manual specification of the emphasized segment or multiple validation runs to identify effective attention heads.

**Failure Analysis of Vision-Language Models**   Our work relates to research on hallucination detection in VLMs.  Chen et al. (2024c) examine multi-object recognition tasks, observing that VLMs exhibit more hallucinations when dealing with multiple objects compared to single-object scenarios. They also note a similar phenomenon to our findings: the distribution of tested object classes impacts hallucination behaviors, suggesting that VLMs may rely on shortcuts and spurious correlations.  Additionally, Tong et al. (2024b) analyze VLM failures from a CLIP perspective, highlighting that the visual capabilities of recent VLMs still face systematic shortcomings, partly due to CLIP's limitations in specific cases.

**Decoding Strategies for Reducing Hallucinations**   This work is also connected to various decoding and tuning strategies aimed at mitigating hallucinations in VLMs.  Leng et al. (2024) introduce a contrastive decoding method that emphasizes certain image regions.  Wang et al. (2024) propose a data-augmentation approach to create image-intensive datasets, followed by preference tuning on this enhanced data. Furthermore, knowledge extraction techniques such as the method proposed by  Chuang et al. (2023) improve decoding by leveraging contrastive layers for better knowledge extraction. Similarly, Activation Decoding (Chen et al., 2024b) identifies optimal answers as those with the highest activation values within the context.

### A.5 CASE STUDY

We show more case we could fix in this section.

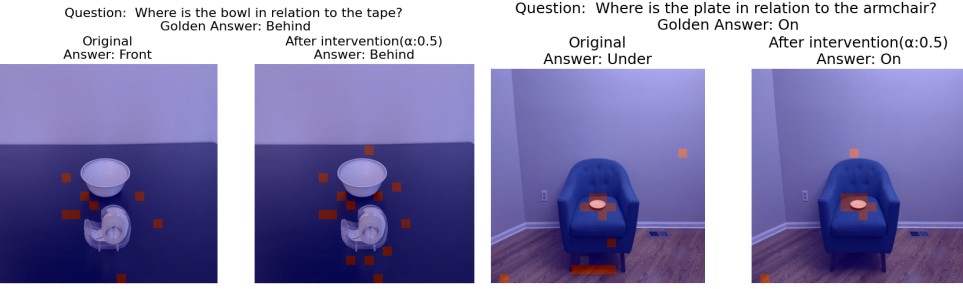

Figure 13: Examples of our fixed case for $\alpha = 0.5$

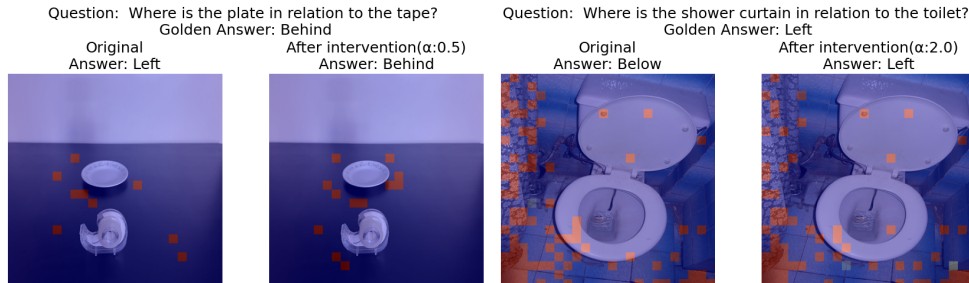

Figure 14: Examples of our fixed case for $\alpha = 2.0$

### A.6 OTHER METRICS TO DISTINGUISH THE DISTRIBUTIONS

After analyzing the experimental results presented in Table 1, we further explored additional metrics to distinguish the distributions and uncover their underlying characteristics during our preliminary experiments. Specifically, we examined entropy and skewness. Entropy was selected based on the hypothesis that parameter differences may stem from the familiarity of attention patterns in real images, which are generally correct, versus synthetic images, where these patterns tend to be incorrect. We posit that the model can express "familiarity" through certain metrics derived from the attention scores. For example, we hypothesize that the entropy of attention will be lower when the model encounters familiar cases.

$$E\left(\mathcal{A}_{n,j}^{(l,h)}\right) = -\sum_{j=1}^{t} \tilde{P}\left(\mathcal{A}_{n,j}^{(l,h)}\right) \log \tilde{P}\left(\mathcal{A}_{n,j}^{(l,h)}\right) \tag{4}$$

In Equation 4, $\mathcal{A}_{n,j}^{(l,h)}$ denotes the attention scores assigned by the $h$-th head in the $l$-th layer to the $j$-th token in sequence $n$. The summation runs over $j = 1$ to $t$, where $t$ is the total number of tokens considered for this attention distribution. $\tilde{P}\left(\mathcal{A}_{n,j}^{(l,h)}\right)$ is the normalized probability distribution of these attention scores. This entropy measures the uncertainty or spread of the attention distribution across tokens.

Our experimental results in Figure A.6 indicate that the attention distribution is heavily influenced by image features. Notably, the attention distribution is more concentrated in synthetic datasets than in real images. We attribute this to the fact that synthetic images tend to contain fewer objects, resulting in a sharper attention distribution. However, this concentration does not provide a reliable metric for measuring familiarity. Another possible metric is skewness. Another possible metric is skewness, which captures the asymmetry of the attention distribution. A high skewness suggests that the attention is predominantly focused on a few positions, while a low skewness indicates a more balanced spread across multiple regions. By examining skewness, we aim to identify whether the attention is being disproportionately allocated to particular image features, which could provide additional insights into how familiarity is expressed through attention patterns. We could see from Figure A.6 that Synthetic datasets show a higher skewness. However, it also related with the object distribution, which is not the real factor behinds the difference.

$$S\left(\mathcal{A}_{n,j}^{(l,h)}\right) = \frac{\sum_{j=1}^{t}\left(j - \mu_{\mathcal{A}}\right)^3 \tilde{P}_j}{\sigma_{\mathcal{A}}^3} \tag{5}$$

The summation runs over $j = 1$ to $t$, where $t$ is the number of tokens considered in the attention distribution. $\tilde{P}_j$ denotes the probability assigned to the $j$-th token in the normalized attention distribution. The term $\mu_{\mathcal{A}}$ is the mean of the distribution, and $\sigma_{\mathcal{A}}$ is its standard deviation. The skewness is calculated as the normalized third central moment, which measures the asymmetry of the attention distribution: a positive value indicates a distribution skewed to the right, while a negative value indicates a skew to the left.

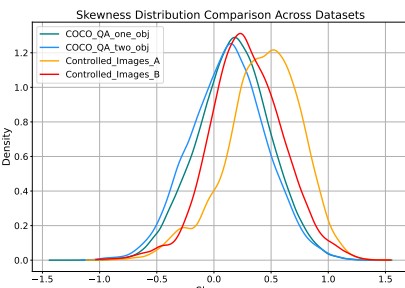 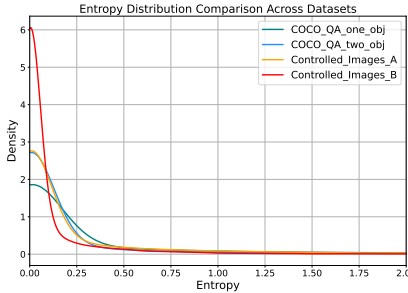

Figure 15: The skewness and entropy distribution comparison between different subsets. Here we use Controlled_Images and COCO datasets due to all of them are in four option label space, which enables us to eliminate the influence of prompts.

| Dataset | Relationship Types | | | | | |
|---|---|---|---|---|---|---|
| | **Right** | **Left** | **On** | **Under** | **Behind** | **Front** |
| Controlled_A | 92 | 92 | 130 | 92 | 0 | 0 |
| Controlled_B | 102 | 102 | 0 | 0 | 102 | 102 |
| VG_one | 376 | 392 | 192 | 198 | 2 | 0 |
| VG_two | 137 | 127 | 3 | 0 | 5 | 19 |
| COCO_one | 564 | 576 | 363 | 744 | 0 | 0 |
| COCO_two | 129 | 150 | 86 | 75 | 0 | 0 |

Table 6: Gold Answer Frequency by Spatial Relation in WhatsUp dataset

## A.7 ADDITIONAL STATISTICS

We present the golden labels' distribution in Table 6. We can see that the label space in synthetic datasets is balanced while the real image datasets are imbalanced with more "left" and "right" and fewer other relationships.

