# OpenReview forum: "AdaptVis: Spatial Understanding in Vision-Language Models Requires Adaptive Attention"
_ICLR.cc/2025/Conference — ICLR 2025 Conference Withdrawn Submission_

### Official Review · Reviewer_HcHm · 2024-10-29

**Soundness:** 2
**Presentation:** 2
**Contribution:** 3
**Rating:** 3
**Confidence:** 4

**Summary:**

This paper introduces ADAPTVIS, a novel adaptive attention mechanism for improving spatial reasoning in vision-language models (VLMs). The approach addresses common spatial reasoning errors by dynamically adjusting the attention distribution on image tokens based on model confidence, thereby enhancing performance in tasks requiring geometric understanding. The authors evaluated ADAPTVIS on benchmarks such as WhatsUp and Visual Spatial Reasoning (VSR), demonstrating substantial accuracy improvements with minimal computational overhead.

**Strengths:**

1. The paper provides a thorough analysis, with visuals offering valuable insights.

2. The method is novel and enhances VLM performance in specific domains.

3. The approach is clearly structured and easy to understand.

**Weaknesses:**

1. The writing needs improvement and re-organization, with more space allocated for experiments. The analysis and visualization sections are overly lengthy and lack strong relevance to the method. Moreover, much of the analysis is based on prior work.

2. The thresholds (alpha1, alpha2, beta) lack generalizability across datasets and VLMs, making them impractical for real-world applications. As shown in Table 4, performance drops significantly compared to Tables 2 and 3, where thresholds were selectively chosen. Suggested solutions: 1) identify optimal thresholds and validate them across more benchmarks/VLMs; 2) create an adaptive algorithm to set thresholds for specific scenarios.

3. Despite the grid search on thresholds in the evaluation set, the performance may still reflect overfitting.

4. Line 399: Figure 10 is incorrectly referenced, and VCD and Dola (in Tables 2 & 3) are unexplained in the main text.

5. Figure 12 is unclear, as it supposedly includes three data types, but only two curves are shown. Additionally, the visualized phenomenon seems overly simplistic.

**Questions:**

1. Which VLM is used in Sections 4 and 5?

2. Has the VLM been fine-tuned on the Controlled Image and VSR datasets?

---

> ### Author Response · Authors · 2024-12-02
>
> Thank you for your constructive review.
>
> **Writing**
>
> Thanks for pointing it out. We have thoroughly revised and proofread the manuscript to enhance clarity and ensure proper definition of all terminology. Our paper focuses on the mechanism interpretability of spatial reasoning. It opens the black box to see why VLMs fail in simple spatial reasoning and how the attention of VLMs behaves under spatial reasoning questions, so we believe mechanism interpretability analysis is of the same importance as the method and provide various analyses in Section 3. We made it clear in the revision.
>
> **Suggestion of creating an adaptive algorithm to set thresholds for specific scenarios.**
>
> We further explored additional metrics in Appendix E to automatically distinguish the distributions and uncover their underlying characteristics during our preliminary experiments. Specifically, we examined entropy and skewness. Entropy was selected based on the hypothesis that parameter differences may stem from the familiarity of attention patterns in real images, which are generally correct, versus synthetic images, where these patterns tend to be incorrect. We posit that the model can express "familiarity" through certain metrics derived from the attention scores. For example, we hypothesize that the entropy of attention will be lower when the model encounters familiar cases.
>
> **Potential Overfitting**
>
> First, the hyperparameter only requires a small number of instances and is more like a calibration. To indicate this, we present the results with models using the same hyperparameter in Table 4, and we can still see consistent improvement. Using the same prompt complexity (all with 4 options), same alpha12 (temperature), and beta (threshold) parameters showed consistent improvements across different benchmarks.
>
> Second, we provide additional results on generic question-answering settings such as GQA, VQAv2, and POPE, as added in Appendix G.7. Our method consistently outperforms the baseline across all benchmarks but with much less gain. It shows that adjusting attention is important and has generalization capability in the general Question Answering setting, but may require different adaptation strategies, which can be interesting future work.
>
> | Dataset     | POPE-A | POPE-P | POPE-R | POPE-AVG | GQA  | VQAv2 |
> |-------------|--------|--------|--------|----------|------|-------|
> | Baseline    | 81.0   | 86.4   | 88.4   | 85.3     | 56.1 | 74.0  |
> | ScalingVis  | 81.8   | 86.6   | 88.6   | 85.6     | 56.3 | 74.2  |

---

### Official Review · Reviewer_QET6 · 2024-11-02

**Soundness:** 3
**Presentation:** 4
**Contribution:** 3
**Rating:** 5
**Confidence:** 4

**Summary:**

This paper focuses on spatial reasoning of large vision language models (LVLMs). Through visualizing the regions of images with the highest attention scores across intermediate layers, the authors notice that errors frequently occur when attention is mistakenly focused on irrelevant parts of the image. Besides, attention patterns vary significantly between familiar spatial relationships (e.g., “on the left side of”) and unfamiliar ones (e.g., “in front of”). The proposed ADAPTVIS adjusts attention based on confidence scores at inference time and performs well on spatial reasoning benchmarks.

**Strengths:**

1. The paper is clearly written and easy to follow.
2. The basic idea of the method is simple but effective for spatial reasoning.

**Weaknesses:**

1. Although the method brings improvement over LLaVA on spatial reasoning benchmarks, the basic idea, sharpening/soothing the
attention when the confidence is high/low is generic for visual-centric understanding. In my opinion, using only spatial reasoning benchmarks is insufficient. Besides, results on common benchmarks (MMBench[a], SEED-Bench[b], etc.) for LVLMs are missing. Whether the ADAPTVIS can improve the common multi-modal capabilities.
2. The datasets used for the main experiments are somewhat simple. The authors should conduct evaluations on GQA [c] or VQAv2, visual question-answering datasets involving spatial reasoning. If using the entire dataset is not suitable, consider using a subset about spatial reasoning instead.
3. Although simple, the ADAPTVIS uses several hyperparameters and the grid search is needed to obtain the values, which increases the complexity.

[a] Liu, Yuan, et al. "Mmbench: Is your multi-modal model an all-around player?." European Conference on Computer Vision. Springer, Cham, 2025.
[b] Li, Bohao, et al. "SEED-Bench: Benchmarking Multimodal Large Language Models." Proceedings of the IEEE/CVF Conference on Computer Vision and Pattern Recognition. 2024.
[c] Hudson, Drew A., and Christopher D. Manning. "Gqa: A new dataset for real-world visual reasoning and compositional question answering." Proceedings of the IEEE/CVF conference on computer vision and pattern recognition. 2019.
[d] Goyal, Yash, et al. "Making the v in vqa matter: Elevating the role of image understanding in visual question answering." Proceedings of the IEEE conference on computer vision and pattern recognition. 2017.

**Questions:**

1. What about the generality of the proposed method? Since the method is applied in the decoding process, it can be evaluated with LVLMs more than the LLaVA series.

---

> ### Author Response · Authors · 2024-12-02
>
> Thank you for the constructive review.
>
> **Only focuses on spatial reasoning**
>
> We appreciate the reviewer’s appreciation and trust in our mechanism interpretability method for generic problems. However, we respectfully disagree that the focus of spatial reasoning is a limitation. It is well-established in the field that mechanism interpretability studies must necessarily focus on specific capabilities or behaviors, as large models exhibit distinct mechanisms across different tasks, different domains, and even different input formats. This targeted approach is not a limitation but rather a methodological necessity, as evidenced by previous successful mechanism interpretability work. We specifically chose spatial reasoning because it represents a critical bottleneck in VLMs' visual understanding capabilities, and our paper presents the first-ever mechanism interpretability work of this important problem. The specificity of our analysis allows us to make precise, actionable discoveries about how VLMs process simple spatial reasoning.
>
> **Additional Datasets**
>
> We provide additional results on generic question-answering settings such as GQA, VQAv2, and POPE. Our method consistently outperforms the baseline across all benchmarks but with much less gain. It shows that adjusting attention is important and has generalization capability in the general Question Answering setting, but may require different adaptation strategies, which can be interesting future work.
>
> | Dataset     | POPE-A | POPE-P | POPE-R | POPE-AVG | GQA  | VQAv2 |
> |-------------|--------|--------|--------|----------|------|-------|
> | Baseline    | 81.0   | 86.4   | 88.4   | 85.3     | 56.1 | 74.0  |
> | ScalingVis  | 81.8   | 86.6   | 88.6   | 85.6     | 56.3 | 74.2  |
>
> **Usage of Hyperparameters**
>
> First, the hyperparameter only requires a small number of instances and is more like a calibration. To indicate this, we present the results with models using the same hyperparameter in Table 4, and we can still see consistent improvement. Using the same prompt complexity (all with 4 options), same alpha12 (temperature), and beta (threshold) parameters showed consistent improvements across different benchmarks.
>
> Second, we further explored additional metrics in Appendix E to automatically distinguish the distributions and uncover their underlying characteristics during our preliminary experiments. Specifically, we examined entropy and skewness. Entropy was selected based on the hypothesis that parameter differences may stem from the familiarity of attention patterns in real images, which are generally correct, versus synthetic images, where these patterns tend to be incorrect. We posit that the model can express "familiarity" through certain metrics derived from the attention scores. For example, we hypothesize that the entropy of attention will be lower when the model encounters familiar cases.

---

### Official Review · Reviewer_JgtL · 2024-11-02

**Soundness:** 2
**Presentation:** 1
**Contribution:** 2
**Rating:** 5
**Confidence:** 4

**Summary:**

The authors explore vision-language models’ struggle with spatial reasoning, focussing on how misdirected attention (i.e. to irrelevant parts of image) within transformer blocks contributes to such behavior. They analyze attention patterns and report how attention prioritizes text tokens over image tokens. They also note that attention to the wrong parts of an image is an issue and how model logit probability can be a proxy for model confidence. Using these ideas, they propose an approach to adjust attention based on confidence levels: sharpening or smoothing the image tokens' attention weights based on model confidence per sample. They evaluate their model on two small datasets with some natural and synthetic images and mostly synthetic captions. The results show promise of their proposed method.

**Strengths:**

1) The authors use of test-time per-sample attention adjustment using model output probability as a proxy for confidence is clever and interesting.
2) Despite the small and mostly synthetic nature of the datasets, the results improvements look promising and add value to verifying the method.
3) The idea of “generation confidence being a reliable indicator of attention correctness” is interesting and serves as an interesting direction for further exploration in the vision-language domain.

**Weaknesses:**

1) Is magnitude of attention score proportional to how much information is used from a token? Could any non-zero attention weight (however low) still correspond to most information within that token being used? The authors seem to assume the reverse at first, and I am unsure if there is any literature to back this up. In fact, later in the paper, the authors actually claim that “location of attention on image tokens is more crucial than quantity”.

2) **Section 3.2 highly unclear**
    1) please explain it better. For Figure 5, the caption nor main text explains the plots clearly.
    2) How is AUROC score calculated?
    3) In Figure 5, what is Cont_A and Cont_B? These are used in Figures throughout the paper but not defined clearly anywhere.
    4) “additional experiment by incrementally increasing the attention weights across
the entire image” - which one in Figure 5 is this?
    5) When referring to Figure 5 in main text, please specify left or right - Figure 5 has two quite different plots in it.


3) **Section 4.1 claims unsubstantiated**
    1) Claim that “the model automatically focuses on the relevant entity when correctly answering questions” is backed by just 4 examples. This 4 image visualization is insufficient to back such a claim.
    2) [Suggestion] Since some of these images are from COCO, bounding box annotations for each image exists. The authors could easily calculate the attention overlap with actual image locations over a larger dataset and provide a metric to measure focus on relevant entity and report correlation between this metric and spatial reasoning accuracy. Without such evidence, this claim remains extremely weak.

4) **Figure 7 unclear and discrepant**
    1) How is the model average confidence calculated? Is this using the ground-truth and averaged for correct predictions only?
    2) What is 'golden' label? Is this ground-truth?
    3) On *COCO_two*: In Table 7 left, COCO_two has non-zero values for *behind* and *front*. However, in Table 6 (appendix), COCO_two has zero such relationships. Please explain this discrepancy and provide more details on what is shown in these plots.
    4) What is the impact of incorrect model predictions on this analysis? This is not discussed at all.

5) L296-L298 typo? Sec 5.1 / 5.2 incorrectly referenced.

6) **About ScalingVis**
    1) In L223-L225, the authors note how “augmenting the image attention logits with a constant coefficient does not improve performance on spatial reasoning tasks.” How is this earlier setup different from ScalingVis? Why does ScalingVis perform different?
    2) The results appear to have the hyper-parameter being tuned specifically for each test dataset. Could the result improvement simply be an over-fitting to each specific test dataset?
    3) Do you have any reasoning / analysis on why sharpening vs smoothing the image attention supports the two synthetic vs real datasets?

7) L395 Equation 5.2 - typo? Please fix.

8) Latex formatting for inverted commas “ ” should follow `` ’’. Bad formatting in L401 and several other places.

9) Baselines in all Tables: please cite and mention in caption and main text what VDC / DoLa are!! In fact, the *dola* name, which I assume is paper [1], is even mis-capitalized.

10) **Limited experimental results**
    1) All results focus on two small datasets where most of the images and possibly all of the captions are synthetic. The generality of this method to more mainstream / real-world tasks remains in question.
    2) The authors follow an evaluation split on these datasets different to prior work (e.g. see [3]) which makes it difficult to compare with prior work. For example, in [4] several works achieve 60+ % accuracy on VSR while the author's baseline achieves 35-40% accuracy. This brings doubts on whether the performance gains are coming from the actual author's method or from simply better parameter tuning. Also it brings doubts on whether the method would improve better performing baselines, especially when those baselines perform almost 50% better than the author's method.
    3) In Table 1, LLaVA 1.6 appears sub-par to LLaVA 1.5 for most dataset splits. One would expect the newer version to perform better. Also, for some splits the baseline achieves 0% or almost 0% performance. Is there a possibility the baselines are not implemented correctly? Especially given how all the reported numbers are significantly lower than numbers reported in prior work.
    4) None of the reported numbers in experimental results tables are from prior works (meaning the authors’ replication could be suboptimal). Especially given how several prior works explore these ideas of spatial reasoning in VLMs (see related work in [2, 5]), it would really strengthen the paper if authors could evaluate their method on a common benchmark used by prior work and compare against numbers reported on those papers.
    5) Minor: Please add details of the dataset splits used for all evaluations. These are not mentioned clearly in the paper.
    6) Minor: Consider including compute used for inference and any changes in timing for implementing the method. From what I gather, the confidence estimate would require one run of the entire network followed by another second run where the attention weights are optimized, leading to at least a 2x slow-down in inference time.


&nbsp;

[1] DoLa: Decoding by Contrasting Layers Improves Factuality in Large Language Models, ICLR 2024

[2] Learning to Localize Objects Improves Spatial Reasoning in Visual-LLMs, CVPR 2024

[3] What’s “up” with vision-language models? Investigating their struggle with spatial reasoning, EMNLP 2023

[4] Liu, Fangyu et al. “Visual Spatial Reasoning.” Transactions of the Association for Computational Linguistics 11 (2022): 635-651.

[5] Hsu, Joy et al. “What's Left? Concept Grounding with Logic-Enhanced Foundation Models.” NeurIPS 2023.

**Questions:**

1) The two main datasets used are from [3] and [4]. Could you report results following the same splits they use so numbers reported are directly comparable to numbers from those papers? This would allow better understanding of how AdaptVis improves over an existing baseline.

2) Could you evaluate this method on a generic VQA dataset like GQA or VQA-v2 and compare against existing prior works? For example, in [1] the authors show that improved spatial reasoning also helps general VQA performance. Maybe AdaptVis will similarly help performance in general VQA tasks too.

3) Several recent works explore and evaluate similar spatial relationships [2,3,4,5]. Applying AdaptVis over baselines from those papers and especially comparing if AdaptVis improves over those works would strengthen the experimental section. Right now, the biggest weakness of the paper appears to be experimental validation of claims.


&nbsp;


Overall I think this is a really interesting direction and idea. However, the current paper is written very poorly, contains several unverified claims, and has a highly flawed experimental setup.


&nbsp;

[1] DoLa: Decoding by Contrasting Layers Improves Factuality in Large Language Models, ICLR 2024

[2] Learning to Localize Objects Improves Spatial Reasoning in Visual-LLMs, CVPR 2024

[3] What’s “up” with vision-language models? Investigating their struggle with spatial reasoning, EMNLP 2023

[4] Liu, Fangyu et al. “Visual Spatial Reasoning.” Transactions of the Association for Computational Linguistics 11 (2022): 635-651.

[5] Hsu, Joy et al. “What's Left? Concept Grounding with Logic-Enhanced Foundation Models.” NeurIPS 2023.

---

> ### Author Response · Authors · 2024-12-02
>
> Thank you for your constructive review.
>
> **Clarity of Section 3.2 and Section 4.1**
>
> In our preliminary experiments, we conducted extensive analyses of attention patterns and their connection to factual errors. We did not include them in the original paper since our observation that "models primarily process information in intermediate layers" aligns with previous findings in [1] and [2]. Additionally, our conclusion that some VLM failures can be attributed to incorrect attention is consistent with the findings in [3]. Due to page limitations, we only include results that show more novel observations in the paper.
>
> The revised version will have additional, comprehensive results in appendices, including an analysis of image attention score values across layers on all subsets of WhatsUp and visualizations of attention areas.
>
> References:
> 1. Overthinking the Truth: Understanding How Language Models Process False Demonstrations, Danny Halawi et al., 2023.
> 2. Dissecting Recall of Factual Associations in Auto-Regressive Language Models, Mor Geva et al., 2024.
> 3. Contrastive Region Guidance: Improving Grounding in Vision-Language Models without Training, David Wan et al., 2024.
>
> **Figure 7**
>
> Thank you for your suggestion. We have clarified the caption to improve its clarity. The average confidence score is calculated as the mean of the model's confidence across each type of question with different ground-truth labels, referred to here as "golden" labels. For COCO_two, we made a typo in Figure 7 (left panel), where COCO2 and VG2 were reversed. This typo has been corrected in the latest version of our work. Regarding the predictions, the pattern aligns closely with the confidence scores, as shown in Figure 10. We observe that the trend in confidence scores corresponds closely to the accuracy scores for each type of ground-truth label.
>
> **About ScalingVis**
>
> In Lines 223–225, the experimental setup involves adding a constant coefficient equally to all image tokens, corresponding to an addition operation in the logit space (equivalent to a multiplication operation in the probability space). In contrast, Scaling Vis uses a multiplication operation directly in the logit space, making it an adaptive approach. Regarding "overfitting," we have shown that our hyperparameter performs well in general settings, as demonstrated in Table 4. As for why sharpening versus smoothing the image attention supports the two datasets (synthetic vs. real), we hypothesize that this is due to differences in their label distributions. Synthetic data contains more unfamiliar relationships, as discussed in Section 4.2, which benefits from a smoother image attention distribution, as observed in Figure 10. In contrast, real images feature more familiar relationships that require a sharper image attention distribution, as shown in Figure 10 as well.
>
> **About Baselines**
>
> We have added the baseline introduction in the latest version of our paper.
>
> **Experimental results**
> - We especially focus on the mechanism interpretability of spatial reasoning and open the black box to see how attention behaves under spatial reasoning questions, since it is well-established in the field that mechanism interpretability studies must necessarily focus on specific capabilities or behaviors, as large models exhibit distinct mechanisms across different tasks, different domains, and even different input formats. This targeted approach is not a limitation but rather a methodological necessity, as evidenced by previous successful mechanism interpretability work. We specifically chose spatial reasoning because it represents a critical bottleneck in VLMs' visual understanding capabilities, and our paper presents the first-ever mechanism interpretability work of this important problem. The specificity of our analysis allows us to make precise, actionable discoveries about how VLMs process simple spatial reasoning.
> - Spatial reasoning especially requires attention on the relative region to reflect the geometric structures, which is a relatively simple and focused attention adaptation, while generic images may have much more complicated attention working mechanisms.
> - We provide additional results on generic question-answering settings such as GQA, VQAv2, and POPE. Our method consistently outperforms the baseline across all benchmarks but with much less gain. It shows that adjusting attention is important and has generalization capability in the general Question Answering setting, but may require different adaptation strategies, which can be interesting future work.
>
> | Dataset     | POPE-A | POPE-P | POPE-R | POPE-AVG | GQA  | VQAv2 |
> |-------------|--------|--------|--------|----------|------|-------|
> | Baseline    | 81.0   | 86.4   | 88.4   | 85.3     | 56.1 | 74.0  |
> | ScalingVis  | 81.8   | 86.6   | 88.6   | 85.6     | 56.3 | 74.2  |

---

> > ### Comment · Reviewer_JgtL · 2024-12-02
> > **Link to updated paper**
> >
> > Could the authors share a link to the updated manuscript including all of the suggested changes? Ideally highlight the changes in a different color. This will be helpful in reviewing the modifications.
> >
> > I cannot see any changes in the current PDF on OpenReview.

---

### Official Review · Reviewer_bgBj · 2024-11-04

**Soundness:** 2
**Presentation:** 2
**Contribution:** 2
**Rating:** 3
**Confidence:** 3

**Summary:**

This paper looks at the weakness of VLMs with spatial reasoning and notions of "left", "right", "above", etc. They do some analysis of the attention scores and argue that image tokens are sparsely attended to, that correct answers are correlated with correct image attention, and that correct answers are also correlated with model confidence. They propose two variants of a method; their main method re-weights attention maps according to confidence. They show some improvement on certain spatial reasoning datasets.

**Strengths:**

They address an important problem which is spatial reasoning in VLMs. The idea of intervening on the attention weights is interesting and has potential, and I think trying to intervene on the mechanics of the model based on internal measures of accuracy like confidence is interesting. They show some improvement on spatial reasoning benchmarks, especially more controlled settings where the label distribution is balanced, which could be promising since VLMs struggle with this task.

**Weaknesses:**

Unfortunately this paper has several weaknesses. The analysis presented in Section 3 and 4 is not comprehensive. For instance, in Sec 3.1 data is only shown for a single dataset. A very strong conclusion is drawn about models primarily processing information in intermediate layers; the empirical evidence is not very strong and again shown on a single dataset. In Section 4.1, there seems to be no quantitative analysis supporting the argument that incorrect answers correlate with incorrect attention patterns; the qualitative examples are also not strong, e.g. the third attention map looks reasonable to me. Overall, Sections 3-4 are overly lengthy but contain insufficient analysis for the strength of arguments made.

I do think the idea of modifying the attention is interesting. However, I am not sure whether I agree with the method conceptually. If the model is confident, it seems reasonable to make the attention sharper -- although if it is confidently wrong, then the attention might be similarly confident and incorrect. If the model is not confident, then diffusing the attention may allow the model to attend to more of the image -- but it seems to me to further reduce the model's confidence. This is especially true since the alpha is applied uniformly across all layers, so the attention is forced to be less focused from beginning to end of the model. So this method seems applicable only when the attention behaves in a certain way (as shown in Fig. 1), but I'm not sure the empirical analysis in Sec 3-4 is strong enough to motivate this approach.

While some of the results on the controlled datasets are promising, I think more time is needed to properly explain the method and improve the empirical analysis. Moreover, the clarity and readability of the paper needs to be improved, e.g. there are several undefined terms used (e.g. Table 2, Dola). The Related work section seems a bit brief but is very hard to understand because it is jargon-heavy and no paper is elaborated on.

**Questions:**

One thing I would encourage is a deeper empirical analysis in Secs. 3-4. It seems to me that your method makes certain assumptions about how attention maps appear in VLMs (indicated by the examples you show in Fig. 1), and I think it would be helpful to prove those more empirically. In particular Section 4.1 could have some more in-depth (and preferably quantitative) analysis of attention patterns. If those sections are clear then this would better motivate the design of the method and help the reader understand why it would work practically.

---

> ### Author Response · Authors · 2024-12-02
>
> We thank the reviewers for appreciating the idea of modifying the attention.
>
> **The analyses in Sections 3 and 4**
> - In our preliminary experiments, we conducted extensive analyses of attention patterns and their connection to factual errors. We did not include them in the original paper since our observation that "models primarily process information in intermediate layers" aligns with previous findings in [1] and [2]. Additionally, our conclusion that some VLM failures can be attributed to incorrect attention is consistent with the findings in [3]. Due to page limitations, we only include results that show more novel observations in the paper.
> - The revised version will include comprehensive results in appendices, including an analysis of image attention score values across layers on all subsets of WhatsUp and visualizations of attention areas.
>
> References
> 1. Overthinking the Truth: Understanding How Language Models Process False Demonstrations, Danny Halawi et al., 2023.
> 2. Dissecting Recall of Factual Associations in Auto-Regressive Language Models, Mor Geva et al., 2024.
> 3. Contrastive Region Guidance: Improving Grounding in Vision-Language Models without Training, David Wan et al., 2024.
>
> **Reliability of Attention Modification**
> - The key idea is to be more focused if the model is confident and widen the focus if the model is unsure. In this case, if the model is not confident, we assume the model should not stick to the original wrong prediction, where “further reducing confidence” is what we want.
> - We especially focus on spatial reasoning and open the black box to see how the attention of VLMs behaves under spatial reasoning questions, since it is well-established in the field that mechanism interpretability studies must necessarily focus on specific capabilities or behaviors, as large models exhibit distinct mechanisms across different tasks, different domains, and even different input formats. This targeted approach is not a limitation but rather a methodological necessity, as evidenced by previous successful mechanism interpretability work. We specifically chose spatial reasoning because it represents a critical bottleneck in VLMs' visual understanding capabilities, and our paper presents the first-ever mechanism interpretability work of this important problem. The specificity of our analysis allows us to make precise, actionable discoveries about how VLMs process simple spatial reasoning.
> - Spatial reasoning especially requires attention on the relative region to reflect the geometric structures, which is a relatively simple and focused attention adaptation, while generic images may have much more complicated attention working mechanisms.
> - We provide additional results on generic question-answering settings such as GQA, VQAv2, and POPE. Our method consistently outperforms the baseline across all benchmarks, but with much less gain. It shows that adjusting attention is important and has generalization capability in the general Question Answering setting, but may require different adaptation strategies, which can be interesting future work.
>
> | Dataset     | POPE-A | POPE-P | POPE-R | POPE-AVG | GQA  | VQAv2 |
> |-------------|--------|--------|--------|----------|------|-------|
> | Baseline    | 81.0   | 86.4   | 88.4   | 85.3     | 56.1 | 74.0  |
> | ScalingVis  | 81.8   | 86.6   | 88.6   | 85.6     | 56.3 | 74.2  |
>
> **Undefined Terms**
>
> Thanks for pointing it out. We have thoroughly revised and proofread the manuscript to enhance clarity and ensure proper definition of all terminology.

---

### Author Response · Authors · 2024-12-02
**General Response**

We thank the reviewers for the suggestions on analysis and experiment setting. We are glad the reviewer finds the idea interesting and the experiment results impressive. Meanwhile, we will revise and improve our submission for the next venue based on the feedback. Here we answer all the questions and hope they can address the primary concerns.

A major concern is whether the spatial reasoning focus of the paper is limited. We especially focus on the mechanism interpretability of spatial reasoning and open the black box to see how the attention of VLMs behaves under spatial reasoning questions, since it is well-established in the field that mechanism interpretability studies must necessarily focus on specific capabilities or behaviors, as large models exhibit distinct mechanisms across different tasks, different domains, and even different input formats. The targeted research of spatial reasoning is not a limitation but rather a methodological necessity, as evidenced by previous successful mechanism interpretability work. We specifically chose spatial reasoning because it represents a critical bottleneck in VLMs' visual understanding capabilities, and our paper presents the first-ever mechanism interpretability work of this important problem. The specificity of our analysis allows us to make precise, actionable discoveries about how VLMs process simple spatial reasoning.

Another concern is whether the analysis of section 3 is comprehensive. In our preliminary experiments, we conducted extensive analyses of attention patterns and their connection to factual errors. We did not include them in the original paper since our observation that "models primarily process information in intermediate layers" aligns with previous findings in [1] and [2]. Additionally, our conclusion that some VLM failures can be attributed to incorrect attention is consistent with the findings in [3]. Due to page limitations, we only include results that show more novel observations in the paper. The revised version will include comprehensive results in appendices, including an analysis of image attention score values across layers on all subsets of WhatsUp and visualizations of attention areas.

We also provide additional results on generic question-answering settings such as GQA, VQAv2, and POPE. Our method consistently outperforms the baseline across all benchmarks but with much less gain. It shows that adjusting attention is important and has generalization capability in the general Question Answering setting, but may require different adaptation strategies, which can be interesting future work.

References
1. Overthinking the Truth: Understanding How Language Models Process False Demonstrations, Danny Halawi et al., 2023.
2. Dissecting Recall of Factual Associations in Auto-Regressive Language Models, Mor Geva et al., 2024.
3. Contrastive Region Guidance: Improving Grounding in Vision-Language Models without Training, David Wan et al., 2024.

---

### Note · Authors · 2024-11-15

I have read and agree with the venue's withdrawal policy on behalf of myself and my co-authors.

---

> ### Note · Program_Chairs · 2024-12-02
>
> **Comment:**
>
> Withdrawal reversed temporarily.
>
> **Revert Withdrawal Confirmation:**
>
> We approve the reversion of withdrawn submission.

---

### Note · Authors · 2024-12-02

**Comment:**

Requested by authors.

**Withdrawal Confirmation:**

I have read and agree with the venue's withdrawal policy on behalf of myself and my co-authors.